# Role of the regulator in enabling a just culture: a qualitative study in mental health and hospital care

Jan-Willem Weenink [1], Iris Wallenburg [1], Laura Hartman [2,3], Eva van Baarle [4,5], Ian Leistikow [1,6], Guy Widdershoven [4], Roland Bal [1]

¹Erasmus School of Health Policy & Management, Erasmus University Rotterdam, Rotterdam, The Netherlands
²Council of Public Health & Society, Den Haag, The Netherlands
³Centre for Ethics and Health, Den Haag, The Netherlands
⁴Department of Medical Ethics, Law and Humanities, Vrije Universiteit Amsterdam, Amsterdam, The Netherlands
⁵Netherlands Defence Academy, Breda, The Netherlands
⁶Dutch Health and Youth Care Inspectorate, Utrecht, The Netherlands

**Correspondence to**
Dr Jan-Willem Weenink;
weenink@eshpm.eur.nl

## ABSTRACT

**Objectives** A just culture is considered a promising way to improve patient safety and working conditions in the healthcare sector, and as such is also of relevance to healthcare regulators who are tasked with monitoring and overseeing quality and safety of care. The objective of the current study is to explore the experiences in healthcare organisations regarding the role of the healthcare inspectorate in enabling a just culture.

**Design** Qualitative study using interviews and focus groups that were transcribed verbatim, and observations of which written reports were made. Transcripts and observation reports were thematically analysed.

**Setting** Three mental healthcare providers, two hospitals and the healthcare inspectorate in the Netherlands.

**Participants** We conducted 61 interviews and 7 focus groups with healthcare professionals, managers and other staff in healthcare organisations and with inspectors. Additionally, 27 observations were conducted in healthcare organisations.

**Results** We identified three themes in our data. First, professionals and managers in healthcare organisations perceive the inspectorate as a potential catalyst for learning processes, for example, as an instigator of investigating incidents thoroughly, yet also as a potential barrier as its presence and procedures limit how open employees feel they can be. Second, a just culture is considered relational and layered, meaning that relationships between different layers within or outside the organisation might hinder or promote a just culture. Finally, for inspectors to enable a just culture requires finding a balance between allowing organisations the time to take responsibility for quality and safety issues, and timely regulatory intervention when healthcare providers are unwilling or unable to act.

**Conclusions** If regulators intend to enable the development of a just culture within healthcare organisations, they must adopt regulatory procedures that support reflection and learning within the organisations they regulate and consider mutual trust as a vital regulatory tool.

## INTRODUCTION

Standards and protocols as well as practices such as root-cause analysis have been instrumental in enhancing quality and safety of care. Increasingly though, criticisms are

## STRENGTHS AND LIMITATIONS OF THIS STUDY

⇒ A strength of this study is the amount and variety of collected data from two healthcare sectors.
⇒ Participating organisations were motivated to work with the inspectorate in this study, whereas including organisations that are less motivated or less comfortable with the inspectorate could have resulted in additional insights.
⇒ The study was conducted in one country, the Netherlands, whereas the precise role of regulation and regulators might depend on the national context of regulation.

voiced about their inability to take into account the complexity of healthcare, urging that further improvements must be sought in culture and behaviour.[1–4] A just culture has been proposed as a means to further enhance quality and safety of healthcare.[5]

The concept of a just culture is not easily described and different meanings and conceptualisations exist in literature and healthcare practice. Reason introduced the concept as an attribute of a safe culture, which has resulted in flow charts or culpability trees to determine whether a healthcare professional should be held accountable for a medical error.[6] Others have highlighted the emotional impact of medical errors and subsequent investigations on healthcare professionals and the need for restorative justice within a just culture.[7–9] A prospective focus on learning and healing is more central in this approach instead of a retrospective focus on understanding the error and whether individuals should be held accountable.[10] Finally, some conceptualise just culture as a culture in which employees feel free to speak up and voice concerns, not only after errors have occurred but whenever they feel the quality of care might be at risk.[11] These conceptualisations are not mutually exclusive and at the same time there are differences in focus and scope. Based on these conceptualisations

though, we could consider openness and dialogue, and balancing accountability and learning and improving, as key characteristics of a just culture.

The concept of just culture—although conceptualised in various ways—has been around for a few decades. Most papers on just culture in healthcare are of conceptual nature.[12–14] Empirical studies about just culture in healthcare remain limited.[15] Those that have been conducted focus on the impact of just culture training on the perceived organisational or safety culture,[16 17] measuring tools for assessing a just culture,[18] what managers need in terms of personal competencies to effectively implement a just culture[19] and on specific aspects of a just culture, such as peer support for second victims.[20]

Because a just culture is expected to contribute to quality and safety of healthcare, the concept is also of relevance to healthcare regulators that are tasked with monitoring and overseeing quality and safety of care.[21 22] The role of regulation has been addressed in just culture literature. Dekker has called for the implementation of just culture in regulatory arenas and internationally there are examples of regulators that have implemented tools to regulate from a just culture perspective.[5 23] Marx noted that 'regulators must become a force for error reduction rather than a force of error concealment'.[24] Little is known however about how regulators impact and could enable a just culture in healthcare organisations. The limited empirical work on just culture focuses mainly on professionals and organisations without considering the impact of the broader healthcare context such as healthcare regulation. The latter could however affect a just culture and initiatives to implement a just culture in healthcare organisations.

The objective of the current study is to explore the role governmental regulation has regarding a just culture in healthcare organisations, and to reflect on what this means for policy and practice of healthcare regulators.

## METHODS
### Setting
Our study focuses on regulation of healthcare in the Netherlands. The role of the Dutch Health and Youth Care Inspectorate (from now on: inspectorate) is to supervise quality and safety of both healthcare organisations and individual healthcare professionals.[25 26] The inspectorate uses two approaches: incident-based supervision following incidents and complaints, and risk-based supervision focusing on specific themes or type of providers. Dutch healthcare organisations are mandated by law to report sentinel events (meaning unintended harm to patients that led to death or serious injury) to the inspectorate and share the investigation report with the inspectorate.[27] In recent years, the inspectorate has focused its policy on learning and improvement of healthcare professionals and organisations, and in this context the current project takes place.

### Study design
Between 2017 and 2019, we studied how five healthcare organisations and a project group of the inspectorate worked on enhancing a just culture in healthcare organisations. The project underlying our study was initiated by the inspectorate with the aim of understanding what is needed for a just culture and how the inspectorate can contribute to this. For the project, researchers conducted a literature review with the objective of developing a working definition of just culture. This definition was not used as a normative framework but as a heuristic instrument to explore our empirical cases. Central elements in the working definition were openness about (a lack of) safety and fallibility, a balance between accountability and learning and improvement, considering different perspectives when an incident occurs, mutual trust between healthcare professionals and in relation to patients, and paying attention to what goes right in addition to what goes wrong. The complete working definition can be found in online supplemental appendix A.

We used qualitative research methods such as observations and interviews to explore experiences with working on a just culture and the relationship with regulation. Participating organisations (ie, three mental healthcare providers and two hospitals) were recruited after a seminar of the inspectorate about just culture and each started their own project on working on a just culture. These projects varied from working on specific processes (eg, incident investigations) to broader approaches on quality and safety policies in the organisation. Simultaneously, a project group of the inspectorate held regular meetings to reflect on preliminary findings and their own role as inspectors in enabling a just culture. The goal of our study was to identify overarching themes related to the role of regulation in enabling a just culture.

### Data collection
In preparation of the empirical research in the organisations, interviews were held with employees of the inspectorate to gain insight into the way inspectors interpret the concept of a just culture and how they view their own role. Subsequently, we observed meetings, held interviews with healthcare professionals, managers and quality and safety officers, and conducted focus group interviews about working on a just culture and the role of regulation in the participating organisations. Topic lists were developed and discussed by the research team to guide data collection. The interviews and focus groups were recorded (audio) and transcribed verbatim, while written reports were made of the observations. During the project, we presented and reflected on preliminary findings within the organisations. In addition, we organised three network meetings. Here, representatives of the five organisations and the inspectorate came together and shared their experiences to learn from each other. At the end of the project, we organised three focus groups with inspectors in which we fed back the results from the organisations and reflected on what these findings mean

**Table 1** Overview of data collection

| Location | Activities for data collection |
|---|---|
| Inspectorate | ▶ 8 interviews with inspectors to explore the concept of just culture and potential role of the inspectorate at the start of the project<br>▶ 3 focus groups with inspectors (4–8 per group) to reflect on the findings at the end of the project |
| Mental healthcare organisation #1 | ▶ 7 dialogue sessions with ±4 participants of different layers of the organisation in which participants discussed experiences and dilemmas in (working on) a just culture<br>▶ 1 reflection session |
| Mental healthcare organisation #2 | ▶ 2 dialogue sessions in which participants discussed experiences and dilemmas in (working on) a just culture<br>▶ 10 interviews with participants of dialogue sessions<br>▶ 6 interviews with professionals<br>▶ 2 interviews with management |
| Mental healthcare organisation #3 | ▶ 17 interviews<br>▶ 4 observations<br>▶ 2 focus groups<br>▶ 1 reflection session |
| Hospital #1 | ▶ 11 interviews with 14 persons<br>▶ 2 focus groups |
| Hospital #2 | ▶ 7 interviews<br>▶ 12 observations<br>▶ 1 conference meeting |
| Network sessions | ▶ 3 meetings with organisations and inspectorate aimed at exchanging experiences between participating organisations |

for the regulator. In total, the data collected for this study consisted of 61 interviews, 7 focus groups and 27 observations. Table 1 provides an overview of all data collection methods.

## Data analysis
The analysis focused on exploring overarching and recurring themes in the data.[28] The first interviews were inductively coded by involved researchers in order to identify common patterns in the data. These patterns included factors and mechanisms related to enabling a just culture in organisations in general, not on the role of regulation specifically. These patterns were then discussed, adjusted and further elaborated on during discussions within the research group, leading to a coding scheme. Subsequent transcripts of interviews and focus groups and notes of observations were coded using this scheme and new themes were added as they emerged. Findings from the transcripts, observations and meetings were discussed within the research group and fed back to participating organisations and inspectors throughout the project. To understand and reflect on the specific role of regulation in enabling a just culture, the findings were further analysed with the purpose of this study in mind. We primarily focused on perceptions and actual experiences with the impact of regulation on a just culture, yet also included respondents' perceptions of the potential role of regulation in enabling a just culture. Our analysis led to three main themes related to regulation and just culture.

## Patient and public involvement
There was no patient or public involvement.

## Findings
From our analysis, we identified three themes that are important to understand the role of governmental regulation in enabling a just culture in healthcare organisations. The first concerns how regulation impacts a just culture in healthcare organisations. The second regards the relational and layered nature of a just culture. This extends beyond the role of regulation alone, yet in this study we focus on how it applies to regulation. The third theme entails specific challenges for regulators and inspectors when trying to enable a just culture in healthcare organisations.

### Regulatory impact on a just culture
When respondents elaborated on the role and impact of the regulator in enabling a just culture, they referred to two important issues: the image of the regulator and the rigidity of forms and procedures.

#### Police or driver of quality improvement?
Respondents perceived the inspectorate as a threat for creating a just culture, as they come into play when things already have gone wrong to judge about what has gone and has been done wrong. Although the inspectorate's scope and tasks are broader, this perception does affect the safety and openness that employees experience when trying to learn from incidents.

> That's how I see the inspectorate. When they come to the hospital, something is going on somewhere. They don't just show up, you know. The police also does not come to your house for a cup of coffee. Then

there is something going on as well. That's how I see it.

There is a perceived threat among professionals of being held responsible for (their share in) an incident, and that being open in their communications can backfire. Inspectors recognised this tension yet referred to the professional standards that healthcare professionals must adhere to. According to inspectors, not calling on individual responsibility is difficult when sentinel events involve culpable personal actions of a professional.

At the same time, the inspectorate may act as an important driver for quality improvement. The image of the inspectorate and possible measures they might take ensure that healthcare organisations take sentinel event investigations seriously. The involvement of the inspectorate makes healthcare organisations want to do such investigations thoroughly and make time and resources available for it.

And well, then we found out that the medication used is already off the market in various hospitals. You just go deeper, deeper and deeper because of that investigation. I just wonder if the mandated investigation [of the inspectorate] had not been there, would we have gone that deep?

The inspectorate's image thus not only has negative consequences, but also implies authority that leads to action in organisations to improve patient safety.

### Rigid forms and procedures when things have gone wrong

Although the inspectorate might be a catalyst for thorough investigative processes, this does not directly mean that these processes also contribute to learning among healthcare professionals. Respondents indicated that the tight timetable with hard deadlines for investigating and reporting sentinel events, in combination with the length of such reports, frustrate openness and thus learning. It means that there is limited space to reflect, and although reflection should be part of the investigation, it is not always experienced as such. There is a risk that meeting the inspectorate's requirements gains priority over the learning process:

There is a time limit of 8 weeks, then it takes time before you get [the report] back from the inspectorate. And you have a time limit in which [a sentinel event] must be reported. So, you cannot just think calmly whether or not to report [the event], whether to investigate.

Respondents experienced that properly recording everything for the investigation is important. They perceive that what is written down on paper is more important than what exactly happened in practice. This 'paper-based reality', in which the focus is mainly on factual matters in combination with the formal language used in the report, is insufficiently in line with how professionals perceive their work. Consequently,

professionals sometimes feel distanced from the reporting.

It is a business-like format that mainly looks at factual summaries of things that have been discussed. (…) People do perceive that [the inspectorate] finds it more important that everything on paper is correct instead of the actual care we provided. Because that story is almost deleted from those documents. (…) So, in the team you see that when the report is finally done, that people have a bit of a hangover from [the investigation report].

Some inspectors acknowledged the need for available instruments and procedures to match the goal of learning and improving and recognise that current forms and procedures do not facilitate such an approach.

I also think that if we want to get a just culture into our DNA from our actions, then we need to carefully examine the systems we work with, the forms we work with and the questions we ask and see whether they are just culture-proof and are focused on learning and prevention in the future.

This requires the inspectorate to reflect on their own procedures from such a learning perspective.

### The relational and layered nature of a just culture

A just culture was considered relational and layered by respondents, meaning that relationships between actors from different layers within and outside the healthcare organisation might hinder or promote a just culture. Two important aspects of this relational and layered nature were mentioned by respondents as relevant for enabling (or hindering) a just culture in organisations: relationships of mutual trust, and the role of publicity and legislation.

### Building relationships of mutual trust

Respondents from the organisations reported that a just culture relies on mutual trust. This applies to different actors within the healthcare organisation, for example, between professionals, superiors and management, as well as in relation to the inspectorate. This means that how employees experience interactions with the inspectorate and individual inspectors—and whether those are positive or negative—is important for experiencing a just culture. Their feeling during inspections is influenced by the procedures and correspondence of the inspectorate.

You wait for some sort of grade from your schoolteacher, it always feels like that. While you would like much more dialogue at the table, 'what do we learn from this'? That the inspectorate thus gains a sense of the learning capacity of an organization from their supervisory role, and not through letters with reference numbers, on which we then disagree and what results in writing another letter back.

Building trust is also about being open about regulatory procedures as an inspector, without perhaps always being able to offer a safe environment. This procedural clarity was seen by inspectors as part of a just culture, in which the quality and safety of patient care must come first. It means that inspectors must be clear that they cannot guarantee that someone will not be held personally accountable in case of sentinel events but that it is very unlikely. Inspectors noticed that this openness and transparency about procedures on their part—in general or specifically when an incident has occurred—contributes to understanding and trust, and respondents from organisations experienced it positively when the inspectorate elaborated and explained their procedures and its position.

> Two inspectors visited the staff and indicated what their working method is and how they view our hospital. And that is quite enlightening. We perceive them as a very annoying organization that is trying to catch us, but in reality, it's not that bad. [Our staff] suddenly sees a face of these people, instead of just their firm notes and letters.

Recognising and building on the relationality of regulation is of importance in enabling a just culture as it helps in building trust and being able to talk about vulnerabilities.

### *The role of publicity and legislation*
A factor that according to respondents influenced openness, and thus learning, is publicity. While openness about an incident or sentinel event within the team or organisation was seen by respondents as an essential component of a just culture, external publication and publicity pose a threat to it. Healthcare professionals perceived that anything they say might become public at some point, either via reports of the inspectorate or via the media. And although the inspectorate does not publish investigation reports with names of professionals, some professionals mentioned that even anonymised data are easily traceable. This perceived threat sometimes leads to openness and learning being disrupted, whereby what is written down on paper is again seen as important. In addition to the perception that paperwork seems more important than learning, choosing what to write down and what not is also about hedging against any potential negative consequences of an investigation report. It could be more worthwhile to sit down and talk and reflect without writing things down on paper, as one of the healthcare professionals said:

> I think so, if we lock ourselves up in a room for one afternoon, then there would just come out more. Things that are not written on paper and that do not go to the [national newspaper]. We would then learn even more from [the sentinel event].

Inspectors recognised the disrupting influence of media and other external actors on a just culture and have to deal with such influences themselves too. They experienced that pressure from the media can lead to a feeling of insecurity among healthcare professionals.

In addition to potential publicity, respondents also mentioned the inhibiting nature of existing legislation and risks of litigation. Even though the inspectorate might adopt a learning perspective in their regulation, existing legislation on disciplinary complaints still focuses on individual accountability. Patients, for example, may choose to file a complaint against individual healthcare professionals.

> There is also the disciplinary judge who is breathing down your neck. We are talking about a just culture, but how open is it when the threat of litigation lingers in the background?

While accounting for sentinel events was thus seen as enabling learning, professionals feared the publicity that might be involved.

### Challenges for regulators and inspectors in enabling a just culture
Respondents, and specifically inspectors, mentioned several challenges for the regulator when trying to contribute to a just culture in healthcare organisations. These challenges related to the assessment of a just culture in practice, and the tension that can arise between informal contact of an inspector with an organisation and formal measures that can be taken.

### *Assessing a just culture*
Inspectors struggled with the question how to assess whether an organisation has a just culture. Some inspectors indicated that instruments are available to get a feel for this, such as inspection frameworks on Trust and on Good Governance, in which openness, transparency and trust are important components. At the same time, inspectors realised that a just culture cannot be ticked off and that it is also about intuition and how confident you are that an organisation itself is able to improve.

> Yes, a gut-feeling. When you are present at the administrative levels, then you need to understand the matter. So you ask the right questions to get a feeling of the organization. If that feeling is not good, then you should take a look at certain indicators.

According to inspectors, asking questions implies a different attitude or style than a controlling one during an inspection visit. At the same time, as an inspector you never only act as a coach and discussion partner.

> We are also assigned a role depending on the situation. So, at one moment it is nice to be a discussion partner and at the other moment an organization needs you as a bogeyman to create urgency.

The inspector cited above emphasises the two roles inspectors can assume and that are expected of them, and the importance of finding a balance between giving space and keeping a firm hand on the healthcare provider. This means an inspector must be able to do both and must

be able to sense which approach is necessary for a given situation.

### Informal contact and formal measures

The fact that the inspectorate can impose sanctions can be at odds with the promotion of a just culture within healthcare organisations. According to inspectors, this makes it difficult because tensions arise between the space that an inspector sometimes wants to give to an organisation to learn and improve (without formal interference from the inspectorate) and the policies and rules that prescribe certain sanctions, such as a monetary fine. Inspectors especially struggled when they had really invested in the relationship with an organisation to ensure that the organisation takes responsibility for quality and safety.

> I am the contact person for this organization. I went through this whole process with them and they clearly learned. It feels wrong to give them a fine at the end of this when they have come up with solutions themselves. I think that in our cooperative relationship, which I understand is a special kind of relationship, that's not good.

Individual inspectors sometimes give a bit of space to organisations because they feel that this contributes to learning within the organisation. Yet, when formally judging an organisation, which is also the inspectorate's task (including possible measures), the case is discussed within the inspectorate. Sometimes there are different perspectives on what should be done between the involved inspector and other inspectors, managers and the legal department. This makes it difficult, because for a healthcare organisation, it might seem as if there is a lot of space to learn and improve and figure things out, whereas at the end of the process the organisation might be confronted with an intervention from the inspectorate. So, although individual inspectors sometimes seem to have an eye for a just culture within the healthcare organisation to facilitate learning, they are aware that trust in the relationship is fragile.

## DISCUSSION

We explored the role of governmental regulation in enabling a just culture in healthcare organisations. Our results show that the regulator, through its procedures and interaction with organisations, has impact on learning processes and openness. Building mutual trust, for example, by being clear about regulatory procedures and expectations, is deemed important, while publicity and external transparency might frustrate learning and openness. Our study moreover highlights challenges for regulators when it comes to assessing a just culture and the impact of legislation. We first provide a brief methodological reflection before we reflect on these findings.

### Strengths and limitations

A strength of this study is the amount and variety of collected data. This rich qualitative dataset from two healthcare sectors enables us to understand processes of just culture and regulation 'from within', which is needed as most of the literature on just culture is theoretical.[7 29] The study has some limitations as well. First, healthcare organisations were recruited after a seminar of the inspectorate and as such were actively interested in working on a just culture with the inspectorate. It is unknown whether studying organisations that are less motivated or less comfortable with the inspectorate would have resulted in additional insights about the role of regulation in enabling a just culture. Second, we need to be careful to generalise these findings across and within settings as the precise role of regulation and regulators might depend on the context of a healthcare sector as well as the national context of regulation. Although the context of regulation will differ internationally, we do believe our findings are of international relevance as the mechanisms we discussed relate to previous findings about the role of regulation.[30 31] How these mechanisms play out in each country might be different and could be input for future comparative research.

### Two issues for regulators when enabling a just culture

For regulators, two issues seem important when aiming to enable a just culture. The first is the impact of regulatory procedures and actions on a just culture in healthcare organisations. As our study showed, the relation between the regulator and healthcare organisations influences the space for openness, reflection and learning in healthcare organisations. It requires reflection from regulators on their policies and procedures, and an understanding of how they directly impact (either positively or negatively) the reflective space in organisations.[2 32] A second issue is just culture as a topic of regulation itself. Inspectors felt the need to be able to assess whether an organisation has a just culture. Although in international contexts tools have been developed to assess a just culture,[33] inspectors indicated that it requires intuition or a gut feeling and that a just culture cannot simply be ticked off. It thus seems important that when choosing an assessment tool, it is used by inspectors to get a better understanding of an organisation by combining it with forms of soft and hard intelligence, instead of directly actioning regulatory measures based on the outcomes of the assessment.[34 35] The latter most likely will not lead to an organisation working towards an open and learning culture but to an organisation trying to score best on the measurements included in the assessment, risking to 'hit the target but miss the point'.[36]

### Balancing conflicting strategies

Regulators are not an independent observer in monitoring quality and safety, but part of the healthcare playing field. Their actions and procedures influence practices within healthcare organisations. In our study,

this was apparent in the mentioned actions of inspectors through coaching or more policing strategies towards healthcare organisations. These strategies are inherent to responsive regulation theory describing persuasive and coercive enforcement approaches, and which assumes that regulators should start with persuasive strategies before considering coercive ones.[37] These strategies conflict and are not strictly successive as theory suggests, for example, when expecting openness from professionals aimed at learning and at the same time keeping the possibility to file a formal complaint against an individual professional. The scope of the Dutch healthcare inspectorate—regulating both organisations and individual professionals—makes this conflict even more complex. At the same time, adopting different strategies makes that healthcare organisations take inspectors seriously, as they can adjust to the specific needs of an organisation. For inspectors, it is important to communicate the intentions of regulation and to continuously reflect on how to balance these strategies in practice.[38] A further question for regulators would then be whether all inspectors should be able to conduct both strategies or whether these styles are represented by different inspectors. The former would require different and for some inspectors new skills, whereas the latter would possibly create tensions between inspectors focused on learning and inspectors focused on policing.[39]

## Beyond the vacuum: taking third parties into account

A just culture requires not only psychological safety in healthcare organisations, but also in the relation between regulator and healthcare organisation.[13 40] However, this relation does not exist in a vacuum, and in enabling a just culture, this is especially problematic when things have gone wrong. Often, patients and patient bodies, politics and media, quite understandably, demand thorough investigations, partly substantiated by a concern that certain things will otherwise be kept under the table. We have seen many examples in the past where these concerns were warranted.[41] The involvement of these other parties also means that the incident and subsequent investigation is taken outside the relation of regulator and healthcare organisation, and the publicity and attention influence openness and learning within the organisation.[42] This is something inspectors are aware of and that poses an additional challenge when trying to enable a just culture as a regulator. Being transparent about regulatory procedures and intentions towards those other parties might contribute to lowering the temperature of heated public discussions and as such contributes to the psychological safety of those involved. For healthcare organisations, directly involving patients or their representatives might contribute to trust and being able to investigate and learn out of the public spotlight.[43]

## Conclusion

Regulators can have an important influence on a just culture in healthcare organisations. This means that when implementing just culture initiatives in healthcare organisations, the role and impact of regulation should be taken into account. For regulators to be able to contribute to a just culture, we recommend that they (1) become aware of the impact regulation and other stakeholders and policies have on a just culture, (2) adopt regulatory procedures that support reflection and learning in organisations, and (3) continuously reflect on how to balance coaching and policing strategies as inspectors. By doing so, regulators can contribute to learning within healthcare and as such improve quality and patient safety.

**Contributors** The study was designed by IW, LH, GW and RB. Data were collected and analysed by JW, IW, LH, EvB, IL, GW and RB. A first draft of the paper was written by JW. All authors contributed to and approved the final manuscript. The guarantor of the study is JW.

**Funding** The study was funded by the Netherlands Organisation for Health Research and Development (ZonMw, project number 516004613).

**Competing interests** None declared.

**Patient and public involvement** Patients and/or the public were not involved in the design, or conduct, or reporting, or dissemination plans of this research.

**Patient consent for publication** Not required.

**Ethics approval** The Medical Research Ethics Committee of the Erasmus Medical Centre determined that the study did not fall within the scope of the Medical Research Involving Human Subjects Act and as such did not require additional ethical approval (MEC-2018-054). All respondents were verbally informed about the study and gave their approval for recording interviews and focus groups. To ensure anonymity, some details from quotations have been adjusted and we do not specify from which organisations quotations come.

**Provenance and peer review** Not commissioned; externally peer reviewed.

**Data availability statement** No data are available. No additional data are available.

**ORCID iDs**
Jan-Willem Weenink http://orcid.org/0000-0003-0443-9785
Iris Wallenburg http://orcid.org/0000-0002-3132-4628
Laura Hartman http://orcid.org/0000-0001-9206-2708
Eva van Baarle http://orcid.org/0000-0003-0212-3090
Ian Leistikow http://orcid.org/0000-0001-6567-0783
Guy Widdershoven http://orcid.org/0000-0001-7620-6812
Roland Bal http://orcid.org/0000-0001-7202-5053

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
