## [Reviewer comments · BMJ Open]

ARTICLE DETAILS

TITLE (PROVISIONAL)	The role of the regulator in enabling a just culture: a qualitative study in mental health and hospital care
AUTHORS	Weenink, J; Wallenburg, Iris; Hartman, Laura; van Baarle, Eva; Leistikow, Ian; Widdershoven, Guy; Bal, Roland

VERSION 1 – REVIEW

REVIEWER	José Mira Miguel Hernandez University of Elche, Health Psychology
REVIEW RETURNED	16-Mar-2022

GENERAL COMMENTS	Thanks for inviting me to review this manuscript. I found it to be a timely, novel and well-conducted study. I believe it should be published. However, I have a few comments that the authors may wish to consider if they feel it would help them improve the paper. In the introduction I believe that the relevance of the proposed objective, including an international perspective, should be more clearly justified. From the reading, an adequate justification of why to carry out this study is not visualized. Although the problem of the definition of just culture is addressed, I believe that the study in the method section should specify what has been considered as just culture in this study and what definition was shared with the participants. In the method section, I believe that it should describe who defined the questions, how the triangulation between the contributions of the different methods of data collection was carried out, and the period of the study, including whether there were other factors that could have had an impact during the study (media impact, pandemic, etc.). The sources of information were diverse. It is not a problem, but it is necessary to provide some detail on how the agreement on the classification of the information was worked out. In my opinion, the discussion would be enriched if some recommendations were addressed and if an international perspective were provided, as there is a wide diversity of approaches, even in Europe.
---

REVIEWER	Kate Churruca Macquarie University, Australian Institute of Health Innovation
REVIEW RETURNED	14-Apr-2022

GENERAL COMMENTS	Thank you for the opportunity to review this manuscript. The study described is an investigation into the perceived role of the Dutch healthcare regulator in a just culture. It is a qualitative study utilizing a large amount of data; the authors should be congratulated for the amount of work they have undertaken in collecting and analysing this. They have provided an interesting and important analysis, with
---

some measured strengths and limitations and significant implications. I have some suggestions for drawing out the analysis further and improving the manuscript.

- Introduction is quite short and while I like the theoretical discussion of the variation and nuances in conceptualizations of just culture a point is made about the limited empirical work on just culture. It would be good to briefly consider this literature, methods, findings etc.
- In terms of the focus of the study, in some sections it's unclear whether the study explores current role of regulator in just culture, or the potential role they could have, or both.
- In the Method, given the potential variation in conceptualization of just culture, it would be good to explain how the regulator defined/positioned it at that initial seminar from which recruitment occurred. It strikes me that the hospitals and regulator could have different views on what constitutes a just culture that would contribute to these findings. Could the authors reflect on this?
- A little more information is needed in the method regarding participants and data collection. What roles from participating hospitals were involved? I note that no participant information is provided with quotes in the Results. Did the authors look for potential differences between participants and sites in their data?
- In table 2 - What are dialogue and reflection sessions as opposed to interviews? Was data generated in these sessions that features in analysis? It would be great to provide example interview questions or better yet the whole interview guide for how you elicited perspectives on just culture. Were interviews conducted in Dutch? If so, could you briefly address the translation process.
- Similarly, I found myself wanting more information on coding and analysis e.g., what types of codes were used and then an example of how they were adjusted. You might also consider this in relation to my next suggestion.
- In the Results, please explain how the subthemes work as part of your three themes.
- Both past and present tense are used in the Results e.g., Respondents emphasize vs. Respondents indicated –I'd recommend past tense.
- The recurrent issue of documentation and paperwork is interesting but I think the authors could unpick it a little more e.g., p. 9 line 4 – is it that the paperwork is factual but partial in its focus? P. 10 line 25 – is this a different almost conflicting issue with paperwork? 1. documentation more important than actual learning 2. documentation as record and a risk for those involved so trying to be selective about what is recorded. Perhaps something for the Discussion?
- P. 13 – the first discussion point on balancing conflicting styles, although interesting, felt a little removed from the findings in the Results. Either reframe or need to make issues of style more explicit in the Results.

Minor

1. In abstract - "observations of which written reports were made" makes it unclear whether the documents were analysed or observations made. Please clarify.
2. P. 8 line 10 – "being open" suggesting adding "in their communications" for clarity.
3. P. 8, lines 49-53 – Is this respondents' view or their perceptions of what the inspectorate prioritizes?
4. P. 8 line 54 – should be "is sufficiently"

	5. P. 9 line 36 – “the feeling that employees get” – I think this could be described more specifically.
--	---

VERSION 1 – AUTHOR RESPONSE

Reviewer 1: Dr. José Mira, Miguel Hernandez University of Elche, Salud Alicante-Sant Joan Health District

Thanks for inviting me to review this manuscript. I found it to be a timely, novel and well-conducted study. I believe it should be published. However, I have a few comments that the authors may wish to consider if they feel it would help them improve the paper. ***Thank you for your compliments and time and effort to provide detailed feedback.***

In the introduction I believe that the relevance of the proposed objective, including an international perspective, should be more clearly justified. From the reading, an adequate justification of why to carry out this study is not visualized. ***In the Introduction on the relevance of the study in the introduction, see also our response to the earlier comment of the editor.***

Although the problem of the definition of just culture is addressed, I believe that the study in the method section should specify what has been considered as just culture in this study and what definition was shared with the participants. ***For the project, we developed a working definition of just culture based on literature review. We have elaborated on this in the Study Design section and have included the working definition as an attachment.***

In the method section, I believe that it should describe who defined the questions, how the triangulation between the contributions of the different methods of data collection was carried out, and the period of the study, including whether there were other factors that could have had an impact during the study (media impact, pandemic, etc.). The sources of information were diverse. It is not a problem, but it is necessary to provide some detail on how the agreement on the classification of the information was worked out.

In the Data Collection section, we've now described who defined the questions and the period of the study. As we did not observe specific other factors (such as Covid, media impact or other events within the organizations) to have impacted the study, we do not make explicit mention of this in the manuscript. In the Data Analysis section, we elaborate on how the first round of interviews was analyzed and led to a coding scheme through rounds of discussion in the research team. This coding scheme was used for analyzing subsequent transcripts of interviews, focus groups and reports of observations, again through rounds of discussion and reflection within the research team.

In my opinion, the discussion would be enriched if some recommendations were addressed and if an international perspective were provided, as there is a wide diversity of approaches, even in Europe.

We have reframed some of our conclusions into more explicit recommendations for regulators in the Conclusion section. In addition, we have now also stated that it is important to take into account the role and impact of regulation when implementing just culture initiatives in healthcare organizations. We've emphasized the international relevance of our study at the

end of the Strengths & Limitations section in the Discussion, yet chose not to elaborate on different approaches of international regulators as we felt this would lengthen the paper too much without adding to the core message of the paper.

Reviewer 2: Dr. Kate Churruca, Macquarie University

Thank you for the opportunity to review this manuscript. The study described is an investigation into the perceived role of the Dutch healthcare regulator in a just culture. It is a qualitative study utilizing a large amount of data; the authors should be congratulated for the amount of work they have undertaken in collecting and analysing this. They have provided an interesting and important analysis, with some measured strengths and limitations and significant implications. I have some suggestions for drawing out the analysis further and improving the manuscript. **Thank you very much for your compliments and taking the time to review this paper and provide such extensive feedback.**

- Introduction is quite short and while I like the theoretical discussion of the variation and nuances in conceptualizations of just culture a point is made about the limited empirical work on just culture. It would be good to briefly consider this literature, methods, findings etc. **We've elaborated more on the empirical work that has been done, see also our previous response to the comment of the editor.**

- In terms of the focus of the study, in some sections it's unclear whether the study explores current role of regulator in just culture, or the potential role they could have, or both. **The focus of our study primarily was on current experiences with regulation and its impact on a just culture, yet respondents also mentioned the potential role the inspectorate could have. We have included both in the paper and have specified this in the Data Analysis section.**

- In the Method, given the potential variation in conceptualization of just culture, it would be good to explain how the regulator defined/positioned it at that initial seminar from which recruitment occurred. It strikes me that the hospitals and regulator could have different views on what constitutes a just culture that would contribute to these findings. Could the authors reflect on this? **The regulator did not use a specific definition of just culture at the symposium; at the start of the project, the research team developed a working definition of just culture, based on literature review. This was shared with the participating healthcare institutions. We have elaborated on this in the Study Design section.**

- A little more information is needed in the method regarding participants and data collection. What roles from participating hospitals were involved? I note that no participant information is provided with quotes in the Results. Did the authors look for potential differences between participants and sites in their data? **We have provided some more information on types of participants in the Data Collection section. We've reported on general and overarching patterns regarding regulation and just culture; no striking differences were observed regarding the role and impact of regulation between sites or participants. Our data showed a consistent image and we therefore**

have decided not to focus on (the lack of) differences between sites and participants in the manuscript.

- In table 2 - What are dialogue and reflection sessions as opposed to interviews? Was data generated in these sessions that features in analysis? It would be great to provide example interview questions or better yet the whole interview guide for how you elicited perspectives on just culture. Were interviews conducted in Dutch? If so, could you briefly address the translation process. **The dialogue sessions refer to group meetings organized by the participating healthcare organization, in which different participants discussed their experiences (e.g. regarding incident investigations). We've added a brief elaboration on this in Table 2. These sessions were in Dutch. The research team observed the sessions and wrote a written report about them that was used as data. Only used quotations from transcripts were translated into English by the authors.**

- Similarly, I found myself wanting more information on coding and analysis e.g., what types of codes were used and then an example of how they were adjusted. You might also consider this in relation to my next suggestion.

- In the Results, please explain how the subthemes work as part of your three themes.

For the project, the data was first analysed on a more general level, focusing on what enables or hinders a just culture in healthcare organizations. This led to a coding scheme with general themes, i.e. communication, speaking up, culture, teamwork and group think, publicity, the role of management, learning and improvement, impact of incidents, and organizational context. For this specific study, this analysis was refined further with a focus on the role of regulation and regulatory work by inspectors, leading to the three main themes and six subthemes. We have not provided additional details about the first step, also because of the earlier comment of the editor about not providing information about the project that is not specifically relevant for this study. We agree that more information is needed on how the three themes and subthemes work, and in the revised manuscript we now introduce each (sub)theme more in detail.

- Both past and present tense are used in the Results e.g., Respondents emphasize vs. Respondents indicated –I'd recommend past tense. **We've changed the text to past tense when referring to experiences of respondents.**

- The recurrent issue of documentation and paperwork is interesting but I think the authors could unpick it a little more e.g., p. 9 line 4 – is it that the paperwork is factual but partial in its focus? P. 10 line 25 – is this a different almost conflicting issue with paperwork? 1. documentation more important than actual learning 2. documentation as record and a risk for those involved so trying to be selective about what is recorded. Perhaps something for the Discussion? **Thank you for this comment; you are right, there are two distinct elements related to paperwork here. We have now specified the distinction in the results section: "In addition to the perception that paperwork seems more important than learning, choosing what to write down and what not is also about hedging against any potential negative consequences of an investigation report." We have chosen not to address this any further in the Discussion as we felt we already address many points in the Discussion.**

- P. 13 – the first discussion point on balancing conflicting styles, although interesting, felt a little removed from the findings in the Results. Either reframe or need to make issues of style more explicit in the Results.

We have reframed this paragraph and moved away from the term ‘styles’ as we do not focus on styles of individual inspectors, but refer to strategies of inspectors and the regulator, as well and the regulator-provider relationship. We do feel this paragraph addresses an important topic in relation to our findings, as it shows the tension between a persuasive and a coercive approach in aiming for learning and improvement and also relates to the perception of the regulator as coach versus police, as part of the first theme.

Minor

1. In abstract - “observations of which written reports were made” makes it unclear whether the documents were analysed or observations made. Please clarify. ***We’ve clarified this by stating that transcripts and observation reports were thematically analysed.***

2. P. 8 line 10 – “being open” suggesting adding “in their communications” for clarity. ***We have changed the text accordingly.***

3. P. 8, lines 49-53 – Is this respondents’ view or their perceptions of what the inspectorate prioritizes? ***This refers to respondents’ experiences and perceptions. They experience that properly recording everything is important and perceive that this is more important than what happens in practice. We have clarified this in the text.***

4. P. 8 line 54 – should be “is sufficiently”. ***We have changed this.***

5. P. 9 line 36 – “the feeling that employees get” – I think this could be described more specifically. ***We have specified this by elaborating that this is about how employees experience interactions with the inspectorate and individual inspectors.***

VERSION 2 – REVIEW

REVIEWER	Kate Churruca Macquarie University, Australian Institute of Health Innovation
REVIEW RETURNED	12-Jul-2022
GENERAL COMMENTS	The authors have addressed my earlier feedback satisfactorily. I have no further concerns about the manuscript. Thank you and well done.